# The Algorithm to Predict the Grain Boundary Diffusion in Non-Dilute Metallic Systems

**DOI:** 10.3390/ma16041431

**Published:** 2023-02-08

**Authors:** Victor Tcherdyntsev, Alexey Rodin

**Affiliations:** Department of Physical Chemistry, National University of Science and Technology “MISIS”, 4, Leninsky pr-t, Moscow 119049, Russia

**Keywords:** grain boundary, diffusion, segregation, interatomic interaction, surface energy

## Abstract

The analysis of grain boundary (GB) diffusion in metallic systems based on Cu, Ni, Ag and Al was made to set demonstrate the common behavior. It was shown that the slow penetration for 11 systems can be connected with negative segregation or specific interatomic interaction. Two energetic parameters such as energy of interaction with GB and energy of interatomic interaction are proposed as main characteristics. The analysis of a tendency toward segregation and tendency of intermediate phase formation in these terms allows us to divide the systems on four groups and formulate a qualitative way to predict the behavior of the diffusing elements in a non-dilute solution. Mathematical formulation of GB diffusion problem and typical solutions are presented.

## 1. Introduction

The prediction of the mass transport in polycrystalline systems is an important problem. It can be solved via a relatively simple way in some cases if diffusion parameters (diffusion coefficient or mobility) are known and the alloy structure is uniform. The term “relatively” here means that we can easily formulate mathematical problem in the case of one-component diffusion, including the cases of self-diffusion or chemical diffusion in a dilute solution. In general, mass transport includes the fluxes of two or more components, and the process is accompanied by vacancy flux, stress, phase formation and other effects. Thermodynamic parameters, mechanical properties and other material characteristics must be incorporated in the model. However, even diffusion characteristics in metals and alloys are known not for all systems [1,2]. In spite of long history of experimental study, the determination of diffusion coefficient is still important and difficult task.

For polycrystalline solids, the problem is more complicated because of presence of grain boundaries (GB) which can be characterized by their own diffusion coefficients. Simple estimation shows that effect of GB on total mass transport must be taken into account if bulk and GB diffusion coefficient ratio approximately equal to ratio between grain size (*a*) and GB width (*δ*) (more than 10^4^–10^5^).

The radiotracer technique is the most sensitive method to obtain the diffusion data, and it is especially useful for GB diffusion study. In this case, the approximation of the dilute solution gets a physical realization because of the possibility of measuring the activity for a very small amount of isotope. It seems that it is a unique method which allows us to directly measure the GB diffusion coefficient *D_b_*. However, for such study, the time–temperature regime must satisfy to absence of bulk diffusion (regime “C” according to Harrison classification [3]), and in practice, it corresponds to very low temperatures—less than 0.5 T_m_. At higher temperatures (0.5–0.7 T_m_), the total flux of certain element in polycrystal includes direct flux from the surface through the lattice (bulk diffusion) with diffusion coefficient *D* and grain boundary diffusion with diffusion coefficient *D_b_*, but because of deeper penetration along GB, the additional flux from GB into grain must be taken into account. It is the so-called “B-regime”. In the simplest case of equality of the GB concentration *C_b_* and concentration in the grain (adjacent to GB zone) *C*, the double product *δD_b_* can be determined, but if *C_b_* and *C* are different, it can be determined by triple product *P = sδD_b_*, where $s={\left(\frac{c_{b}}{c}\right)}_{x=\pm \delta /2}$ (*x* is coordinate normal to GB plane) [4,5].

The characteristics of GB diffusion are less known than the of bulk diffusion [1,6], and they are mostly presented as a double or triple product. Most part of these data were obtained by radiotracer method. However, for practically important cases, the concentration of the diffusing element is much higher. Can we predict their behavior? Let us start with analysis of already-known results.

### Diffusion Behavior in “B” and “C” Regimes

In Figure 1, the grain boundary diffusion coefficient of different elements in the same matrix (Cu) [7,8], obtained by the same method, are presented.

Note that here, the temperature dependencies were extrapolated to high temperatures. Thus, the picture can be divided on two parts: low-temperature part (1000/T > 1.5) where the measurements were mainly made; high-temperature part (1000/T < 1.45), which is the typical temperature range for the “B” regime. Such extrapolation corresponds to the assumption of constant activation energy, which typically means no phase transition in GB, the same diffusion mechanism for given elements, etc. In the low-temperature part, the huge difference in the *D_b_* value can be seen, while extrapolation gives a rather small difference for the elements except Co, Fe and partly Ni. In Figure 2, the data for triple product *P* are presented for diffusion in Cu (Ag, Ni, Bi, Ge, Co, Fe, Cu, Au, Zn, In, As) [9,10,11,12,13,14,15,16,17,18,19], Ni (In, Au, Ag, Sn, Nd, Ce, Ni [20,21,22,23,24,25]), Al (Ga, Ge, Zn, Fe, Cu, Al [26,27,28,29,30]) and Ag (Se, Te, Ni, Ag [31,32,33]). The results are presented for the temperature range 0.5–0.7 T_m_ (for each matrix), and the huge scatter (with factor 10^3^÷10^4^) can be seen for each matrix. For example, the value of triple product for Bi in Cu is 10^3^ times higher than for self-diffusion, while the extrapolation of *D_b_* gives the difference within order of magnitude.

This effect is good illustration of influence of segregation (adsorption). The equilibrium phenomenon of segregation can be defined as enrichment of the surface (interface) in comparison with volume by some certain substances. Formally enrichment coefficient *s*, introduced above, is not the same as in equilibrium case, but typically, it can demonstrate the tendency. Thus, the diffusing elements can be divided on segregated and not segregated ones.

It should be mentioned that compilation of GB diffusion data obtained in the “B” and “C” regimes allows us to obtain segregation factor s from diffusion measurements only. According to the approach in [7], the extrapolation of *p*-values on the low-temperature range gives sδ=PDb. The results of such a calculation presented in [7] gives a variation range for *s* value of 10 ÷ 10^5^. The lowest values correspond to the systems with complete solubility (Au in Cu), and the highest correspond to the systems with small solubility (Bi in Cu or Te and Ni in Ag).

The main effect of segregation on GB diffusion is the change in flux from GB to the grain bulk. The high value of enrichment coefficient means a small concentration of the diffusant in the grain zone, adjacent to GB, and the flux in the direction perpendicular to the GB plane decreases proportionally to this concentration.

Presented in Figure 2 are data which demonstrate that *p* values for self-diffusion are the lowest.

The exceptions are Ni and Au in Cu and Cu and Fe in Al. We can also mention that Co in Cu has almost the same value of *p* as Ni and Cu. It is important to recall that Co, Fe and Ni were mentioned as slow diffusing elements in GBs of Cu. Recent data on Co [34] and Fe [35,36,37] diffusion in polycrystalline Cu at a high concentration level (up to 4–5 wt.%) show the absence of accelerated GB diffusion for these elements in pure Cu, as well as in Cu-Fe and Cu-Co alloys. On the contrary, a radiotracer study showed that preliminary alloying of Cu by Fe (in concentration of 0.8 wt.%) increase *p* value for Fe diffusion on 10^3^ times at a very high temperature [38]. In fact, these facts provoke the necessity for current discussion and the importance of building the algorithm to analyze the results of GB diffusion.

Keeping in mind that the diffusion data mentioned above were obtained on polycrystalline pure materials, we can ignore the effect of the GB structure, e.g., that of the misorientation angle. In this case, we can discuss the problems in terms of effect of interaction of the diffusant with GB of matrix (GB segregation) and of chemical interaction between elements.

## 2. Results and Discussion

### 2.1. Analysis of Segregation Factors

The situation with segregation is more complicated. According to reviews of GB segregation [39,40] the main part of spectroscopic data was obtained on Fe. Additionally, we have some data about GB segregation in Cu and Ni. It is connected with the fact that the experiments can be made only if we can obtain the open surface for analysis, and thus, we must obtain a brittle GB fracture. Naturally, such a brittle fracture can be achieved for elements which segregate on GB and weaken the interatomic bonds.

The analysis of the data gives the simple correlation between the solute concentration at solubility limit, expressed in atomic fraction (*C*_0_), and the GB equilibrium enrichment factor *s*
(1)$$s\approx \frac{\exp\left(-\frac{10\pm 6\mathrm{kJ}/\mathrm{mol}}{RT}\right)}{C_{0}}$$

*C*_0_ is less than unity, and thus, the *s* value is greater than unity for all cases (positive segregation). The value in the bracket has a physical meaning of standard Gibbs energy of segregation (with the opposite sign), averaged for the chosen systems. That is the so-called Hondros and Seah’s rule [39]. More detailed analysis can be found, e.g., in [41].

If we now compare the value of the triple product (Figure 2) with the prediction from Equation (1), we can see that the elements with the highest value of *P* also tend to segregate at GB.

We can add some direct evidence about Ni segregation in Cu, which, according to most studies (see the brief review in [42]), is negative.

Another type of systematic study is the measurement of surface and GB energy by the zero creep method, combined with thermal etching [43,44,45,46]. For the systems on the base of Cu, the surface and GB energies isotherms were obtained in the form:(2)$$\sigma =\sigma_{1}-ZRTln\left(1+bX_{2}\right)$$

Here, *b* is thermodynamic parameter corresponding to equilibrium constant of segregation process. This approach generally confirmed Hondros and Seah’s rule, but besides this, it demonstrates that for In, Sb, Sn, and Bi in Cu, the Langmuir–McLean isotherm [47] can be applied in order to describe the concentration dependence of segregation (segregation isotherm):(3)$$X_{2b}=\frac{bX_{2}}{1-X_{2}+bX_{2}}$$

Using Equation (2), we can take into account that in the dilute solution, segregation and surface tension is connected as:(4)$$c_{ib}=-\frac{c_{i}}{RT}\frac{\partial \sigma }{\partial c_{i}}$$

The isotherm (3) can be obtained with the replacing of *b* in Equation (2) to (*b* − 1) in Equation (3).

Note that *b* and *s* can be compared for the dilute solution in the approximation of quasi-equilibrium between the grain boundary and adjacent grain.

The experiments with Cu-Fe [46], Cu-Co [48] and Cu-Ag [49] systems in the Cu-rich zone demonstrate some anomalies: the surface and GB energy isotherms have a maximum in the case of Co and Fe and a minimum for Ag. Such behavior corresponds to the change in the sign of segregation according to the Gibbs Equation (4).

We can conclude that zone of small concentration of Co and Fe corresponds to negative segregation with factor *s* < 1. Positive and negative segregation means a different case of interaction between the diffusant and grain boundary.

According to Burton [50,51], the estimation of tendency to positive or negative segregation can be made by analyzing of solidus and liquidus lines on the phase diagram. This approach allowed us to make the same conclusion about negative segregation of Ni, Fe, and Co in Cu and positive for all other elements in chosen systems.

### 2.2. Chemical Interaction Factors

The full description of interaction even in two component system is quite a difficult task. Thermodynamic assessment with correct prediction of equilibrium between different phases is based on building Gibbs energy as a function of temperature and concentration. To simplify it, we take into account the main characteristics of binary phase diagram in order to obtain a qualitative estimation of the parameters: solubility in solid and liquid states, and tendency to form intermediate phases (chemical compounds). Here, the crystalline state corresponds to the grain bulk, and the liquid state is an approximation of the non-crystalline structure corresponding to GB. Interatomic interaction can be described in terms of regular solution approximation, and it can be determined as a difference between interaction of atoms of different types and the atoms of the same type. In this way, we pass from an atomistic description to a macroscopic one.

It Is clear that the case of complete solubility both in liquid and solid states corresponds to a weak interaction. Restricted solubility of the components can be connected with two factors:tendency to solution decomposition due to the positive energy of interactions, with the formation of two different solution based on different elements;tendency to the formation of intermediate phase due to negative energy of interaction between components.

In both cases, the higher the absolute value of interaction energy, the lower the solubility. In addition, the intermediate phase forming due to strong interatomic interaction must be stoichiometric and characterized by the narrow range of its stability. Using this approach, we can discuss the difference in the system using only one energetic parameter, and for a two-component system, it can be estimated from the phase diagram, which is known.

### 2.3. Tendency to Association

GB segregation models, developed recently [52,53,54], include the idea that in the system with a strong interatomic interaction, the GB solution can be considered as a solution with associates where segregated elements exist not only in the form of free atoms moving in GB, but also in the form of some complexes or associates consisting of two or more atoms which are relatively stable in time (the life-time must be much longer than the characteristic time of the atomic jump).

In very general form, the process can be written as a chemical reaction in the grain boundary:(5)$${\mathrm{mA}}_{b}+{\mathrm{nB}}_{b}={\left(A_{m}B_{n}\right)}_{b}$$

Index ‘*b*’ indicates that components are at GB. The equilibrium state between GB and the bulk can be described with two equilibrium constants for the atomic exchange between grain boundary and the adjacent bulk (Langmuir–McLean isotherm [47]):(6)$$b=\frac{a_{A}a_{Bb}^{}}{a_{Ab}^{}a_{B}}=\frac{C_{A}C_{Bb}^{}}{C_{Ab}^{}C_{B}}$$
and for the given reaction:(7)$$K_{}=\frac{a_{\left(A_{m}B_{n}\right)b}}{{\left(a_{Ab}^{}\right)}^{m}{\left(a_{Bb}^{}\right)}^{n}}=\frac{C_{\left(A_{m}B_{n}\right)b}}{{\left(C_{Ab}^{}\right)}^{m}{\left(C_{Bb}^{}\right)}^{n}}$$
where $a_{i}$ and $C_{i}$ are, respectively, the activity and concentration of the *i*- component in the grain, and with index *b*, correspond to GB characteristics.

Additionally, if we express the concentration as a molar fraction, we can write $X_{A}+X_{B}=1,$
$X_{Ab}+X_{Bb}+X_{(A_{m}B_{n})b}^{}=1$, or for the total concentration of the diffusing element on GB:(8)$$X_{Bb\Sigma }=X_{Bb}+\frac{n}{n+m}X_{(A_{m}B_{n})b}^{}$$

Formally, the parameters of this model are the ratio m/n, *b* and *K*. It can be supposed that m and n correspond to a possible stoichiometric intermediate phase in this system.

Naturally, this approach is valid for the system with a strong tendency toward chemical compound formation (negative interaction and small solubility). However, it can also be applied to the case of positive interaction if the atoms of the segregating element tend to form couples of atoms or more complicated structures. For example, it was confirmed in our computer simulation [55] for different values of energy of interaction between atoms where the B_n_ complexes were observed. In this case, we just can put m = 0.

### 2.4. Analysis of the System

Coming back to the diffusion results, we can analyze the system using the criteria mentioned above.The systems with complete solubility: Cu-Ni, Cu-Au, Ni-Au. Due to the increase in liquidus and solidus temperature with increasing Ni concentration, the negative Ni segregation in Cu is predicted to be opposite to the case of Au segregation in Cu and Ni, where it must be positive or zero.The system with restricted solubility without chemical compounds. The compounds can be of eutectic or peritectic type. According to the Burton prediction, the eutectic type of the phase diagram corresponds to a tendency toward positive segregation: Cu-Ag, Cu-Bi, Ni-Ag, Al-Ge, Al-Zn, Al-Ga; meanwhile, the peritectic type corresponds to a tendency toward negative segregation: Cu-Co and Cu-Fe.The system with restricted and small solubility and with chemical compounds which have a strong tendency toward forming chemical compounds: Ni-Nd, Ni-Ce, Al-Cu, Al-Fe, Ag-Te, Ag-Se. The last two systems are also characterized by phase decomposition in a liquid state.Other systems. They are characterized by restricted but significant solubility and, thus, can be described as intermediate between the 2nd and 3rd groups.

The data necessary to classification are summarized in Table 1.

### 2.5. Application to Grain Boundary Diffusion—Algorithm

Negative segregation. The first step must be to understand the interaction of the diffusant with GB. As mentioned above, we can divide the system into three groups, with positive segregation, negative segregation and weak segregation effect.

In [56], it was suggested to rewrite the GB diffusion equation for Fisher’s [4,5] model with an additional term associated with the surface energy gradient:(9)$$\frac{\partial C_{b}}{\partial t}=D_{b}div\left\{\frac{\partial C_{b}}{\partial z}-\frac{C_{b}f}{RT}\frac{\partial \sigma }{\partial z}\right\}-\frac{2}{\delta }D{\left.j_{y}\right|}_{y=\frac{\delta }{2}}$$

One can see that segregation parameters here play a role in two terms: the second term (in figure brackets), where it determines the surface energy concentration dependence, and in the last one, where it determines the flux from GB into the bulk.

If the surface energy isotherm is linear in some concentration range, then
(10)$$\frac{\partial \sigma }{\partial z}=\frac{\partial \sigma }{\partial C_{b}}\frac{\partial C_{b}}{\partial z}=-\left(\frac{s-1}{s}\right)\delta RT\frac{\partial C_{b}}{\partial z}\;\mathrm{and}\;\frac{\partial C_{b}}{\partial t}=D_{b}div\left\{\frac{\partial C_{b}}{\partial z}+C_{b}\Omega \left(\frac{s-1}{s}\right)\frac{\partial C_{b}}{\partial z}\right\}-\frac{2}{\delta }D{\left.j_{y}\right|}_{y=\frac{\delta }{2}}$$

It is clear that if $C_{b}\to 0$, term $C_{b}\Omega \left(\frac{s-1}{s}\right)\frac{\partial C_{b}}{\partial z}\to 0$. So, this term is important if the boundary concentration is not very small.

If *s* > 1 the term $\left(\frac{s-1}{s}\right)<1$ and positive. Calculations made in [54] showed that for s > 1, this additional term does not play a role because an increase in *s* leads to a decrease in $\frac{\partial C_{b}}{\partial z}$ (the longer the concentration profile, the deeper the GB diffusion). However, if *s* < 1, this term becomes negative and can be more than unity by absolute value. At the same time, the lower the *s* value, the lower *C_b_* will be.

Linear dependence of surface energy also means the linear segregation isotherm, and using an approximation of the steady state diffusion regime on GB Fisher’s equation for GB diffusion, it can be rewritten as:(11)$$\left\{\frac{\partial^{2}C_{b}}{\partial z^{2}}+\frac{\partial }{\partial z}\left(C_{b}\Omega \left(\frac{s-1}{s}\right)\frac{\partial C_{b}}{\partial z}\right)\right\}-\frac{2\sqrt{D}C_{b}}{s\delta D_{b}\sqrt{\pi t}}=0$$

The solution of this equation for boundary condition *C_b_*(*z* = 0) = *C_b_*_0_ and *C_b_*(*z* = ∞) = 0.
(12)$$z=-\int_{C_{b0}}^{Cb}\frac{L\left(1+A\xi \right)}{\xi {\left(1+\frac{2A\xi }{3}\right)}^{1/2}}d\xi$$
where $A=\Omega \left(\frac{s-1}{s}\right)$ and $L_{F}^{2}=\frac{s\delta D_{b}\sqrt{\pi t}}{2\sqrt{D}}$.

It can be compared (Figure 3) with the usual Fisher–Gibbs solution for the equation
(13)$$c_{b}\left(z,t\right)=c_{0}\cdot \exp\left(-\frac{z}{L_{F}^{}}\right)$$

The main effect is that GB diffusion will be less pronounced (in comparison with the zero-segregation case), not only because *s* < 1 but because of the effect of an additional driving force. The constant enrichment coefficient taken in this calculation means that concentration profile in the bulk (typically measured in the experiment) will be exactly the same. The difference between the curves depends not only on the s value but also on the *C_b_*_0_ value. We can see that far from the initial surface, the lines start to be parallel to one another and effect of surface energy gradient disappear.

It is Important to remember that the main motivation to develop this model was the absence of accelerated GB diffusion of Fe and Co in Cu [34,35,36,37]. This model perfectly describes the result, taking for estimation an enrichment factor of 0.3, which can also be obtained from the data [46,49].

Positive segregation. If we assume the positive segregation, the effect described above is very small. According to the Fisher–Gibbs model, the effect of segregation directly changes the flux from GB into the grain, and this can be seen in the last term of Equation (9), where we must express *C_v_* as a function of *C_b_*.
(14)$$C_{v}=F\left[C\right]\;\mathrm{or}\;C=F^{-1}\left[C_{b}\right]$$

In the simplest case of linear segregation, $C=s^{-1}C_{b}$.

To simplify, the quasi-stationary regime with boundary condition corresponding constant surface concentration will be taken:(15)$$\left\{\begin{array}{l} \frac{\partial^{2}C_{b}}{\partial z^{2}}=L_{F}{}^{-2}F^{-1}\left[c_{b}\right]\\ c_{b}\left(0,t\right)=c_{b0}\\ c_{b}\left(\infty ,t\right)=0 \end{array}\right.$$

Thus, the task was solved for different cases with the use of different isotherms (Langmuir–McLean, Fowler, Temkin) [57,58]. For example, for the Langmuir–McLean isotherm, the expression *F*^−1^ is:(16)$$F^{-1}\left[C_{b}\right]=\frac{C_{b}}{b\left(1-\left[1-1/b\right]C_{b}\right)}$$

Taking into account that all types of isotherms at small concentrations can be presented as linear, the effect may be important for the case of significant GB concentration. The main effect is that for the initial part of the concentration profile, the measured value of concentration will be less than predicted.

According to Table 1, the segregation of most parts of the elements can be described according to the Langmuir–McLean isotherm, and in a very dilute solution, the measured triple product values must be in good correlation with the enrichment factor, albeit taking into account the difference in *D_b_*. A combination of data in Table 1 and Figure 1 and Figure 2 gives a reasonable prediction. However, some cases clearly need to be explained, such as Fe and Cu in Al, Ce and Nd in Ni, and Ni in Ag.

The first four systems are good candidates for the application of the complex formation model. Taking into account the phase diagram, the Al_3_Fe, Al_2_Cu, Ni_5_Nd and Ni_5_Ce can be taken as candidates for the possible complexes. As for the last one, it is an example of the system with phase separation in both liquid and solid states, and the one-component complex formation (Ni_2_, Ni_3_, … Ni_n_) can be suggested.

The difference in the GB diffusion process description is that only free atoms of the diffusant must be taken into account for GB flux, but both free atoms and complexes participate in atomic exchange with the grain bulk.
(17)$$\frac{\partial C_{b\Sigma }}{\partial t}=D_{b}\frac{\partial^{2}C_{b}}{\partial z^{2}}+{\left.\frac{2}{\delta }D\frac{\partial C}{\partial y}\right|}_{y=\partial /2}$$ here, *C_b_* is the concentration of GB free atoms, and $C_{b\Sigma }$ is the total concentration of the diffusing element. In the quasi-stationary regime, the right part is equal to zero, but the second term of the left part requires us to take into account all participants of the GB solution.
(18)$$\left\{\begin{array}{l} \frac{\partial^{2}C_{b}}{\partial z^{2}}-\frac{2\sqrt{\pi Dt}}{\delta D_{\Gamma З}}F^{-1}\left(C_{b\Sigma }\right)=0\\ C_{b}\left(0,t\right)=C_{b0}\\ C_{b}\left(\infty ,t\right)=0 \end{array}\right.$$

Considering the expression for equilibrium constants *b* and *K*, function *F*^−1^ can finally be expressed using the current $C_{b}$ value.

For example, for a dilute solution of Cu in Al GB with Al_2_Cu complexes, one can obtain:(19)$$F^{-1}\left[c_{b}\right]=\frac{1}{b}\cdot \frac{C_{b}\left(z,t\right)}{1-KC_{b}\left(z,t\right)}$$

Additionally, the solution of the equation is:(20)$$z\left(t\right)=-\int_{c_{b0}}^{C_{b}}\frac{d\xi }{\sqrt{-\frac{2}{bK^{2}L{\left(t\right)}^{2}}\left(K\xi +\ln\left|1-K\xi \right|\right)}}$$

A comparison of the bulk concentration profiles for different values of *K* with the same *b* is presented in Figure 4.

For the given parameters, one can see a significant change in diffusion penetration length. For each type of complex different function $F^{-1}\left[c_{b}\right]$ must be used, but the result will be the same. GB diffusion will be slower than in the case of simple atomic segregation.

This developed approach allowed us to predict the behavior of different metallic elements during GB diffusion in another metals. It seems possible to use this algorithm for nonmetallic elements, e.g., O, S, P, C, N. Taking into account that for the diffusion study of these elements, the radiotracer method is typically used, and the diffusant concentration is extremely small, these data were not taken into account.

## 3. Conclusions

Based on the grain boundary diffusion experimental results analysis, the idea of an algorithm to predict the behavior of the diffusing elements in a non-dilute solution is formulated.

In the analysis, the macroscopic approach was developed, and an atomistic description was used only to demonstrate the physical meaning of the effect. According to that, we try to use the characteristics which can be determined in macroscopic studies, such as diffusion coefficients, segregation factor, energy of mixing and energy of segregation. It is also known that diffusion and segregation parameters depend dramatically on the GB structure, but taking into account that in polycrystals, the fraction of special GBs is quite small (around 10% according to [59]), and the problems of GB structure and atomic mechanisms of GB diffusion are out of the discussion.

In order to simplify the mathematics, some important effects (e.g., concentration dependence of diffusion coefficient, grain boundary migration, etc.) were also not taken into account. In addition, for all solutions, the approximation of quasi-stationary regime were applied.

The basic parameters which must be known are the GB diffusion coefficient (or the GB diffusion triple product) obtained by radiotracer techniques and thermodynamic parameters or (as minimum) the phase diagram.

The key points of the algorithm are:Analysis of tendency toward segregation (positive or negative).Analysis of tendency toward complex formation.

In the case of positive segregation without a complex formation, the extended Fisher–Gibbs model can be used for calculation.

In the case of positive segregation with a complex formation, the concentration profile should be built taking into account that not all atoms of diffusing elements will participate in the GB diffusion process. The algorithm to construct the mathematical problem is suggested.

In the case of negative segregation, the additional driving force connected with surface energy gradient should be taken into account.

All factors slowing down the grain boundary diffusion play important an role if the GB concentration of the diffusant is significant.

## Figures and Tables

**Figure 1 materials-16-01431-f001:**
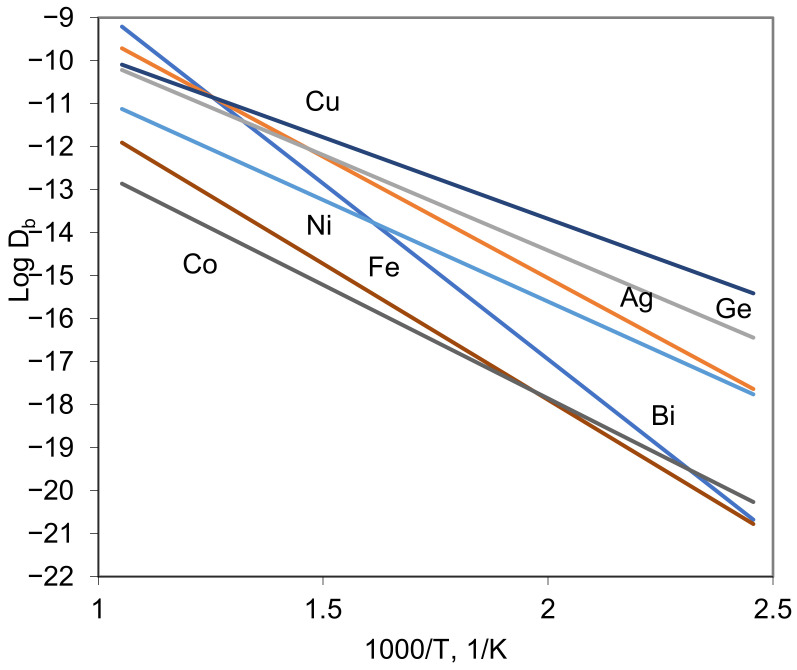
Compilation of recent results on Ag, Ni, Bi, Ge, Fe, Co and Cu GB diffusion in 5N8 high-purity Cu measured in “C” regime [9,10,11,12,13,14,15].

**Figure 2 materials-16-01431-f002:**
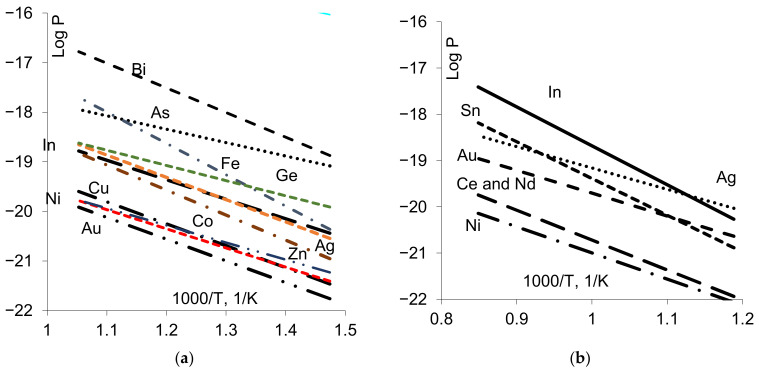
Compilation of the GB diffusion data in Cu (**a**), Ni (**b**), Al (**c**) and Ag (**d**) for “B” regime.

**Figure 3 materials-16-01431-f003:**
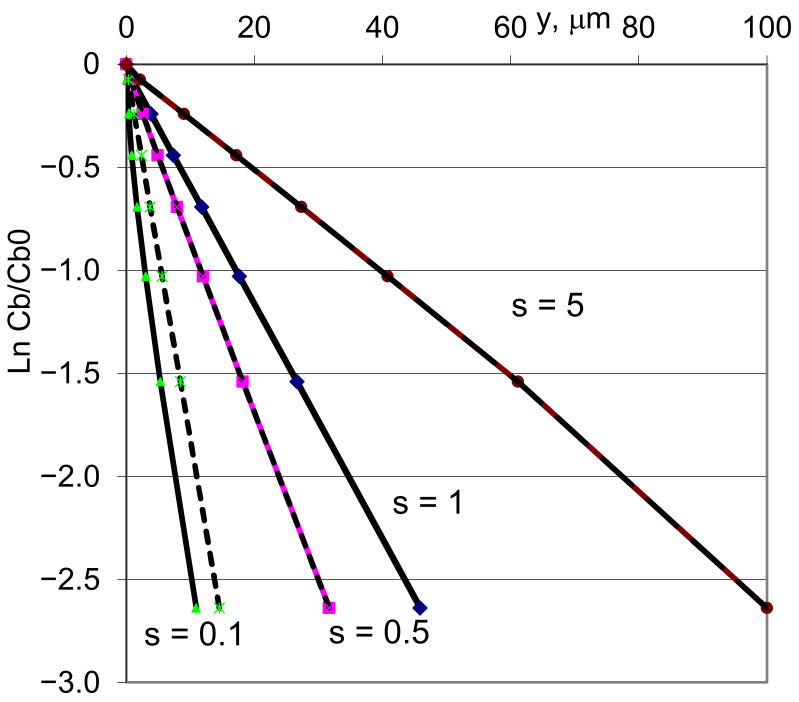
Modelling of GB diffusion profile with diffusion parameters corresponding to Cu self-diffusion at 550 °C, *C_b_*_0_ = 0.1. Enrichment coefficient is variable parameter (according [56]).

**Figure 4 materials-16-01431-f004:**
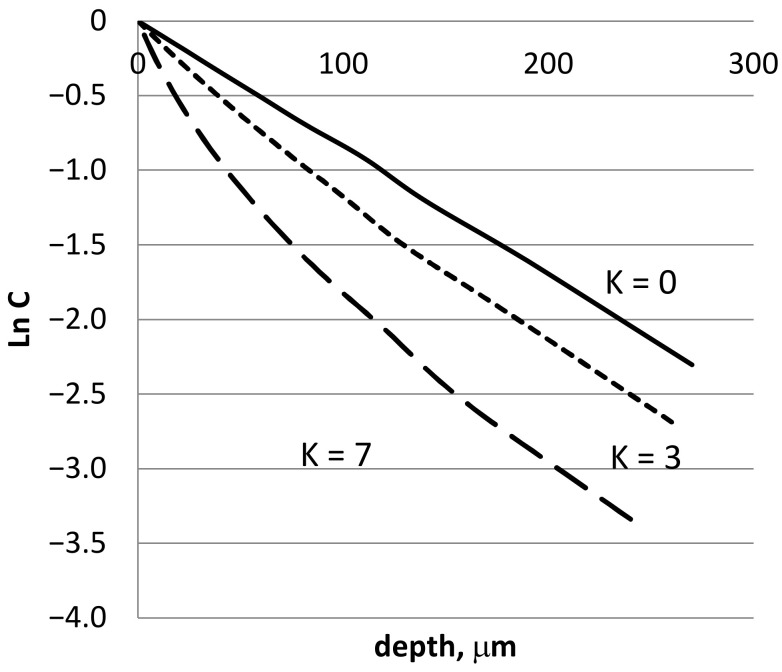
Modelling of GB diffusion profile for bulk concentration near GB in the case of A_2_B complex formation. Diffusion parameters correspond to Zn diffusion in Al at 350 °C, and *b* = 5.

**Table 1 materials-16-01431-t001:** Description of interaction between elements and between GB and diffusing elements (El. is name of element, *C_s_* is maximal solubility, ph.f is tendency to phase formation, assoc. is tendency toward atomic complex formation; negative or positive segregation is denoted by ‘−’ or ‘+’).

El.	*C_s_* (at%)/ph.f	Liquidus/Segregation	Assoc.	El.	*C_s_* (at%)/ph.f	Liquidus/Segregation	Assoc.
Cu matrix							
Au	100/No	Decreasing/weak ‘+’	No	Zn	38/weak	Decreasing/weak ‘+’	No or weak
Ni	100/No	Increasing/weak ‘−’	No	Fe	3	Increasing/High ‘−’	‘+’
Ag	5/No	Decreasing/High ‘+’	No	Co	5	Increasing/High ‘+’	‘+’
Bi	<0.01/No	Decreasing/High ‘+’	No	As	7/Yes	Decreasing/‘+’	‘+’
Ni matrix							
Au	100/No	Decreasing/weak ‘+’	No	Ce	1	Decreasing/High ‘+’	‘+’
In	18/weak	Decreasing/‘+’	Weak	Nd	0.05	Decreasing/High ‘+’	‘+’
Sn	11/weak	Decreasing/‘+’	Weak				
Al matrix							
Ga	9/No	Decreasing/weak ‘+’	No	Fe	0.05/Yes	Decreasing/High ‘+’	‘+’
Ge	4/No	Decreasing/weak ‘+’	No	Cu	4/Yes	Decreasing/High ‘+’	‘+’
Zn	60/No	Decreasing/weak ‘+’	No				
Ag matrix							
Se	0.1/yes	Decreasing/weak ‘+’	No	Ni	0.1/No	Decreasing/‘+’	‘+’
Te	0.1/yes	Decreasing/weak ‘+’	No				

## Data Availability

Not applicable.

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
