# Peer review of "The Algorithm to Predict the Grain Boundary Diffusion in Non-Dilute Metallic Systems"

_materials, 2023, doi:10.3390/ma16041431_

Round 1
Reviewer 1 Report
The scientific content of this ms. is referring to the use of an algorithm to predict the grain boundary diffusion in non-dilute metallic systems. The study is well-organized and examines the different aspects of the systems. The main key-points of the algorithm are that with this approach the analysis of tendency to segregation (positive or negative) and the analysis of tendency of complex formation could be accomplished. The results and the discussion are satisfactory, while some minor points could be addressed by the authors.
1. The authors declare the significance of the approach followed, however a comparison of their results with the existing approaches for the analysis of binary systems is lacking. (e.g. see Materials Transactions, Vol. 44, No. 1 (2003) pp. 14 to 27).
2. The grain boundary diffusion and segregation results obtained from the radiotracer measurements compared to the present analysis should be elaborated in more detail.
3. The English language should be improved throughout the manuscript. The text should be revised by a native English speaker. In addition, there are numerous phrases within manuscript with typos or grammatical errors that need to be revised. For instance, line 42-44, page 1, “and in practice it is corresponds” need to be revised as “and in practice it corresponds”.
Author Response
Thank You very much for Your review. Here are our comments:
- Completely agree.
- We tried to add some analysis. In fact the all our ideas based on the analysis of these data.
- We check it again. A lot of errors was found
Reviewer 2 Report
Reviewer’s comments on “The Algorithm to Predict the Grain Boundary Diffusion in Non-Dilute Metallic Systems.”
This manuscript presents a mathematical formalism to predict the behavior of diffusing elements in non-dilute solutions. The author provides a detailed numerical interpolation of the GB diffusion for elements such as Cu, Ni, Ag, and Al. The work is important for a better understanding and assessment of such a complex problem and is worth publishing in this journal.
However, some revisions are necessary for the publication.
(1)
The authors should discuss the accuracy of the extrapolation to high temperatures (1000/T<1.45), called B region?!
(2)
What are G and A terms in Fig. 2(a)?!
(3)
What is 10+-6 term eq(1)? And why it is different from the “delta_Gibbs free energy” of the solute, adopted in the original Hondros and Seah’s equation.
(4)
The authors should pay attention to the variable names, cite correct authors' names, and provide the correct reference numbers.
Here, I have mentioned some examples:
- C0 is not the solubility limit of the solute. Instead, it is the solute concentration at the solid solubility limit, expressed in atomic fraction.
- Author name: “Burton” instead of “Barton.”
- Reference citation: “Hondros and Seah’s rule [39]” instead of “Hondros and Seah’s rule [40]”.
(5)
The work is lacking discussion of the effect of vacancies and dislocation mobility on the GB segregation. Therefore, the authors are invited to extend their discussion and conclusion, if possible, to include dislocations and vacancy concentration effects in their work.
(6)
Overall, the manuscript requires careful native English editing before consideration for publication.
Error examples:
-Missing commas in the abstract’s first sentence.
-“Positive and Negative segregation” should be discussed in independent subtitles.
-Some incomplete sentences are throughout the manuscript.
- Missing equation number in “equation ()”, page 9, line 256.
Edited abstract to consider: “The analysis of GB diffusion in metallic systems on the base of Cu, Ni, Ag, and Al was made to set demonstrate the common behavior. It was shown that the slow penetration of the 11 elements could relate to negative segregation or specific interatomic interaction. Two energetic parameters (energy of interaction with GB and energy of interatomic interaction) are proposed as main characteristics. Analysis of the tendency to segregation and tendency of intermediate phase formation in these terms allows the possibility to divide the systems into four groups and formulate a qualitative way to predict the behavior of the diffusing elements in a non-dilute solution. A mathematical formulation of the GB diffusion problem and typical solutions are presented”.
Author Response
Thank You very much for Your review. Here You can find some our comments
(1)- Done
(2) just a mistake due to size change of the picture Ge and Ag
(3) It is numerical characteritics, and it can be desribed as “delta_Gibbs free energy” but averaged for different systems. Some comments added to the text.
(4) Agree.
(5) Here we avoid to discuss these effects. Firstly- because lack of the data, and beqacase of additional difficulties in interpretation. If we discuss the vacancies appeared due to the atomic flux differences, we even do not know how to formulate mathematical problems. Or, more exactly, the mathematical difficulties start to be dominating. If it concerns vacancies and dislocation as a driving force for segregation and acceleration of diffsuion near GB.... We think it is another problem. Really important, but required another description and mainly on atomic level.
(6) Completely agree and tried to do our best.
Reviewer 3 Report
On the base of analysis of experimental results on grain boundary diffusion the idea of algorithm building in order to predict the behavior of the diffusing elements in non-dilute solution is formulated. The analysis of GB diffusion in metallic systems on the base of Cu. Ni, Ag, Al was made in order to set demonstrate the common behavior. This work is interesting and meaningful. This work has important implications for predicting the diffusion problem study of polycrystalline materials. But there are still some questions that need to be answered.
1. line 21 “Prediction of the mass transport in polycrystalline systems is important problem” should be revised into “Prediction of the mass transport in polycrystalline systems is an important problem”. There are many similar grammatical errors, please check and modify them one by one.
2. The difference between the lines in Figure 1 and Figure 2 is not obvious enough, so it is suggested to combine different colors and different formats for display.
3. The influence of the grain boundary on the diffusion is very complex and diverse. How to consider this problem? Whether can the formula in this paper fully reflect the diffusion problem at the grain boundary?
4. How is the interaction between the atoms at the grain boundary considered?
Author Response
Thank You veru much for review. We tried to make all the correction and improvement You suggested. Besides we tried to clarify the description in the text. Here we add some small remerks.
- Done
- done
- here we suggested to use Fisher geometry of GB with some modification of driving forces, but still keeping some simplification (e.g. constant GB diffusion coefficient, dominating the diffusion of 1 component, etc.) - eq 9.
- We tried to introduce it as energy of mixing.